# Clinically Relevant Pancreatic Fistula after Pancreaticoduodenectomy: How We Do It

**DOI:** 10.3390/biology12020178

**Published:** 2023-01-22

**Authors:** Jana Enderes, Christiane Pillny, Hanno Matthaei, Steffen Manekeller, Jörg C. Kalff, Tim R. Glowka

**Affiliations:** Department of Surgery, University Hospital Bonn, 53127 Bonn, Germany

**Keywords:** pancreaticoduodenectomy, Whipple, pancreatic fistula, endoscopic vacuum-assisted therapy

## Abstract

**Simple Summary:**

Pancreatic cancer is one of the most aggressive solid tumors with a very poor prognosis. The only opportunity for long-term survival is surgical resection in combination with chemotherapy. Pancreatic surgery, however, is a highly complex and technical challenging procedure, even if carried out by very experienced surgeons. One of the most feared complications after pancreatic surgery is leakage of pancreatic fluid into the abdominal cavity, where it can lead to other serious complications, such as abscess formation, impaired gastric motility and severe bleeding with subsequent death. The goal of this study was to investigate possible risk factors for the leakage of pancreatic fluid and to analyze possible treatment strategies. Our data show that patients developing leakage of pancreatic fluid were younger, mainly male, with fewer comorbidities and with a higher body mass index. Also, they had a smaller tumor size and softer pancreatic parenchyma. In general, these patients showed a worse outcome leading to a prolonged hospital stay; however, this did not affect the overall mortality rate. Treatment strategies included conservative treatment, drainage placement, endoscopic negative pressure therapy and surgery. The majority of the patients were able to receive conservative treatment, resulting in a shorter length of their hospital stay.

**Abstract:**

(1) Background: This study’s goals were to investigate possible risk factors for clinically relevant postoperative pancreatic fistula (POPF) grade B/C according to the updated definitions of the International Study Group of Pancreatic Surgery and to analyze possible treatment strategies; (2) Methods: Between 2017 and 2021, 200 patients were analyzed regarding the development of POPF grade B/C with an emphasis on postoperative outcome and treatment strategies; (3) Results: POPF grade B/C was observed in 39 patients (19.5%). These patients were younger, mainly male, had fewer comorbidities and showed a higher body mass index. Also, they had lower CA-19 levels, a smaller tumor size and softer pancreatic parenchyma. They experienced a worse outcome without affecting the overall mortality rate (10% vs. 6%, *p* = 0.481), however, this lead to a prolonged postoperative stay (28 (32–36) d vs. 20 (15–28) d, *p* ≤ 0.001). The majority of patients with POPF grade B/C were able to receive conservative treatment, followed by drainage placement, endoscopic vacuum-assisted therapy (EVT) and surgery. Conservative treatment resulted in a shorter length of the postoperative stay (24 (22–28) d vs. 34 (26–43) d, *p* = 0.012); (4) Conclusions: Patients developing POPF grade B/C had a worse outcome; however, this did not affect the overall mortality rate. The majority of the patients were able to receive conservative treatment, resulting in a shorter length of their hospital stay.

## 1. Introduction

Pancreatoduodenectomy (PD) is a highly complex and technically challenging procedure. When carried out by certified pancreatic surgeons at high-volume centers, mortality is as low as 6% [1]; however, morbidity still ranges from 30 to 50% [2] with postoperative pancreatic fistula (POPF) being one of the most dreaded complications, often leading to other complications such as intra-abdominal abscess formation, postpancreatectomy haemorrhage and delayed gastric emptying [3,4]. 

According to the International Study Group of Pancreatic Surgery (ISGPS), pancreatic fistula is defined as drainage output on or after postoperative day three with amylase levels greater than three times the normal level of serum amylase [4]. After an update of these definitions in 2016 postoperative pancreatic fistula is now further classified as either an asymptomatic biochemical leak without a change in clinical management (biochemical leak; BL) or as a symptomatic clinically relevant POPF (grade B/C POPF), with grade B POPF requiring prolonged drainage over 3 weeks or additional percutaneous or transgastric drainage and grade C POPF being associated with organ failure or requiring reoperation [5]. With these updated and more precise definitions, the high incidence of POPF ranging from 24.4 to 34.5% [6,7] could be lowered to 11.1–22.0% including only clinically relevant POPF grade B/C [1,7,8,9]. However, POPF still remains the leading cause of the overall high morbidity after PD and thus, we wanted to first critically evaluate the possible risk factors for POPF grade B/C according to the update of the ISGPS definitions and, second, analyze possible treatment strategies. 

## 2. Materials and Methods

Between January 2017 and January 2021, all patients that underwent PD at our department (*n* = 200) were retrospectively studied and prospectively recorded in a pancreatic resection database with the approval of the institutional ethics committee (ethics committee of the Rheinische Friedrich-Wilhelms University Bonn, 347/13) and after obtaining written informed consent from the participants. Patients developing clinically relevant POPF grade B/C were compared against patients not developing clinically relevant POPF grade B/C and analyzed in relation to demographic factors, comorbidities, intraoperative characteristics, hospital stay, morbidity and mortality and postoperative complications. Clinically relevant POPF grade B/C was defined according to the update of the ISGPS definitions from 2016 as amylase level > 3 times the normal serum amylase with a change in postoperative management: for POPF grade B with intraoperatively administered drains either left in place > 3 weeks or requirement of additional percutaneous or transgastric drain placement, or for POPF grade C with the need for surgery or occurrence of organ failure [5]. PPH and DGE were also classified according to the appropriate ISGPS definitions [2,10], and morbidity and mortality were documented according to the Dindo–Clavien classification [11]. 

PD was performed by 3 certified senior pancreatic surgeons (JCK, SM, TRG). Resection and reconstruction via duodenoenterostomy, pancreatogastrostomy and end-to-end choledochojejunostomy were carried out by default in a standardized fashion and as previously described [12,13,14]. Standardized or extended lymphadenectomy defined by the ISGPS definitions [15] was performed depending on the assumed histology, and a sample of the gall fluid was taken by default and sent for microbiological analysis. 

Perioperative management was carried out according to our institutional standard enhanced recovery after surgery (ERAS) protocol as described previously [16]. In brief: prior to surgery patients received additional enteral sip feeds or parenteral nutrition in case of weight loss. Bowel preparations were not carried out. Patients were allowed solid food and liquids 6 and 2 h before surgery, respectively. Directly after surgery patients were allowed to drink water or tea, and, only if amylase levels were normal on postoperative day (POD) 3, patients were allowed an easily digestible/fat-reduced diet followed by an easily digestible/fiber-reduced diet on POD 4, a basic diet (no pulses/no brassica) on POD 5 and a normal diet on POD 6. In the case of vomiting, transition to a normal diet was discontinued and an NGT was re-inserted. Amylase levels in abdominal drains were measured on postoperative day 3 by default. In the case of development of clinically relevant POPF grade B/C, patients were either treated (a) conservatively by food restriction, parenteral nutrition for 7 days, administration of antibiotics and successive removal of intraoperatively administered soft drains; (b) by additional drainage placement in case of intra-abdominal fluid collections; (c) by additional endoscopic vacuum-assisted closure therapy (EVT) in case of insufficiency of PG; (d) by surgery; or (e) by a combination of the aforementioned strategies. If amylase levels on POD 3 were normal, intraoperatively administered soft drains were removed, and intraoperatively administered NGT was removed if secretions were less than 500 mL per day.

Data were recorded and analyzed with Excel 2013 (Microsoft Corporation, Redmond, WA, USA) and SPSS 28 (IBM Corporation, Armonk, NY, USA). Statistical analyses were carried out as described in one of our previous studies [12]: continuously and normally distributed variables were expressed as means ± standard deviation and analyzed using Student’s *t* test, while non-normally distributed data were expressed as medians and interquartile range and analyzed using the Mann–Whitney U test. Categorical data were expressed as proportions and compared with the Pearson χ^2^ or Fisher’s exact test as appropriate. Factors with *p* ≤ 0.1 in the univariate analysis were included in multivariate stepwise logistic regression analysis with a significance level of *p* ≤ 0.05 for inclusion and *p* ≤ 0.10 for removal in each step. The relative risk was described by the estimated odds ratio with 95% confidence intervals. A *p*-value ≤ 0.05 was considered statistically significant.

## 3. Results

Of the 200 patients undergoing pancreatoduodenectomy, 39 patients (19.5%) developed clinically relevant POPF grade B/C. These patients were younger (65 (53–73) years vs. 69 (60–76) years, *p* = 0.025), mainly male (74% male vs. 26% female, *p* = 0.021) and had fewer comorbidities (2 (1–3) vs. 2 (2,3), *p* = 0.014) compared to patients not having developed POPF grade B/C. Also, they showed a higher body mass index (27.0 (24.7–31.2) kg/m^2^ vs. 22.5 (22.0–28.4) kg/m^2^, *p* = 0.002) and accordingly we observed less preoperative weight loss amongst them (33% vs. 58%, *p* = 0.010) (Table 1). Preoperatively, they showed lower levels of CA 19-9 (14.5 (7.1–49.5) and received less biliary drainage, and, interestingly, cholangitis occurred equally in both groups (8% vs. 9%, *p* = 1.000). Intraoperative data such as surgery duration, blood loss and erythrocyte concentrates being transfused did not differ among the groups (Table 1). As for tumor characteristics, patients developing POPF grade B/C less often showed malignant tumor histology (49% vs. 82%, *p* ≤ 0.001) and had softer pancreas parenchyma (72% vs. 40%, *p* ≤ 0.001). Also, tumors of patients developing POPF grade B/C were smaller in size (2.4 (1.2–2.9) cm vs. 3.0 (2.0–3.9) cm, *p* = 0.018) (Table 1). 

Development of POPF grade B/C was associated with more complications in general, a more frequent rate of reoperations (33% vs. 14%, *p* = 0.004) and a worse outcome according to the Clavien–Dindo Classification (Clavien grade III/IV 85% vs. 43%, *p* ≤ 0.001) (Table 2). Interestingly, occurrence of PPH grade B/C (33% vs. 26%, *p* = 314) was not increased in patients developing POPF grade B/C; however, as suspected, DGE grade B/C was more often observed amongst them (46% vs. 17%, *p* ≤ 0.001), including ISGPS-related parameters such as first day of solid food intake, duration of intraoperative administered nasogastric tube (NGT), rate of re-insertion of an NGT or the need for parenteral nutrition. POPF grade B/C not only led to a longer stay in the ICU (3 (1–4) d vs. 1 (1–3) d, *p* = 0.057) but also to a longer postoperative stay in general (28 (22–36) d vs. 20 (15–28) d, *p* ≤ 0.001). Importantly, the overall mortality rate did not differ between the two groups (10% vs. 6%, *p* = 0.481).

In the univariate analysis, the following factors qualified for multivariate analysis: preoperative biliary drainage, positive intraoperative microbiology, intra-abdominal abscess formation, PPH grade B/C and insufficiency of DE. Importantly, POPF grade B/C was not a risk factor for an increased mortality (*p* = 0.481); however, in the multivariate analysis, PPH grade B/C (*p* = 0.007) and intra-abdominal abscess formation (*p* = 0.013) were independent risk factors for a higher mortality rate (Table 3).

Next, we assessed management strategies of POPF grade B/C: the majority of the patients (15 patients, 38.5%) were able to receive conservative treatment (Figure 1 and Appendix A) with food restriction and parenteral nutrition for 7 days, administration of octreotide as well as antibiotics and successive removal of intraoperatively administered soft drains. Conservative treatment resulted in a shorter length of postoperative stay (24 (22–28) d vs. 34 (26–43) d, *p* = 0.012) and a subsequent less frequently positive MTL30 index (7% vs. 46%, *p* = 0.013) compared to surgically or interventionally treated patients.

Nine patients (23.1%) needed additional drainage due to intra-abdominal fluid collections (Figure 1). Of these, in five patients, CT-guided drainage was sufficient to drain the fluid collection, three patients needed additional transgastric drainage and one patient received transgastric drainage in the first place due to cellulitis of the abdominal wall (Appendix A). Six patients (15.4%) with POPF grade B/C qualified exclusively for an endoscopic vacuum-assisted closure therapy (EVT) (Figure 1) with insufficiency of PG between 4 and 10 mm requiring two to seven cycles of therapy until closure of the insufficiency was reached; however, in one case completion pancreatectomy due to severe bleeding was necessary (Appendix A). Three patients (7.7%) with POPF grade B/C exclusively required surgery (Figure 1). In one case this was due to abnormal drainage fluid (suspicious for small bowel leakage, not confirmed), in a second case due to a dehiscence of the abdominal wound and in a third case due to arrosion bleeding of the common hepatic artery. In these three cases the operations comprised lavage with splenectomy, exclusive lavage and completion pancreatectomy with splenectomy, respectively (Appendix A). The remaining six patients needed combined treatment of the above-mentioned treatment strategies (Figure 1 and Appendix A).

## 4. Discussion

The fairly high incidence of POPF ranges from 24.4 to 34.5% in randomized controlled trials [6,7] and is mainly due to the asymptomatic grade A POPF, which is why many studies, including the present study, only focus on clinically relevant grade B/C POPFs that involve a change in postoperative management. The incidence of 19.5% of these POPFs in the present study is comparable to the current literature reporting incidences between 11.1 and 22.0% [1,7,8]. Furthermore, the amylase content in abdominal drains is determined by default on POD 3. However, POPF is still the leading cause of the overall high morbidity after PD and has remained the most dreadful complication [5]. 

Over the last years, specific risk scores have been developed [17,18,19], with the latest, the alternate fistula risk score, identifying BMI, soft pancreatic gland tissue and duct diameter < 5 mm as potential risk factors for POPF grade B/C. In our cohort, in general, patients developing grade B/C POPF showed fewer comorbidities by having a lower CCI. However, we observed a higher BMI and accordingly less preoperative weight loss in patients developing grade B/C POPF, which is in line with previous reports [20,21]. A possible reason could be a higher amount of intrapancreatic fatty tissue in patients having a higher BMI since intrapancreatic fat was shown to be proportional to BMI [22] and thus leading to a softer pancreatic texture, which in turn is known to be one of the most significant risk factors for the development of POPF [23,24], as this study also demonstrates. Accordingly, hard pancreatic tissue with pancreatic fibrosis was shown to be associated with a lower incidence of POPF [25]. It is known that fibrotic pancreatic tissue develops as a result of activation of pancreatic stellate cells and subsequent deposit of extracellular matrix and is observed in both pancreatic cancer and pancreatitis [26], and in mice it was shown that Carbohydrate Antigen (CA) 19-9 induced not only pancreatic cancer but also pancreatitis by activation of epidermal growth factor [27], which is also known to regulate pancreatic fibrosis [28]. Just recently it was shown that preoperative high serum levels of CA 19-9 were associated with a lower risk for development of a pancreatic fistula after PD due to CA-19-9-induced fibrosis [9]. In accordance with this, in our study, levels of CA 19-9 were also higher in the group that did not develop clinically relevant POPF and cholangitis, another possible reason for elevated CA 19-9 [29], occurred equally in both groups. Interestingly, even though we did not observe a difference in the occurrence of cholangitis between the two groups, patients developing grade B/C POPF preoperatively received less biliary drainage, as reported in other studies [9,21], and in multivariate analyses preoperative biliary drainage was even considered to be a protective factor for POPF [21]. It is well known that preoperative biliary stenting is associated with an increased rate of complications as shown in randomized controlled trials [30,31], though manipulation of pancreatic parenchyma as seen in stent placement might also lead to locally inflamed pancreatic tissue promoting fibrosis, which in turn is known to be associated with a lower incidence of POPF [25]. Interestingly, patients developing grade B/C POPF showed a smaller tumor size, as also seen in a recently published study [9] in which small T1 and T2 tumors were associated with higher rates of grade B/C POPF. The degree of healthy non-fibrotic pancreatic tissue might also account for this observation since pancreatic cancer is associated with the induction of fibrosis [26]. With smaller tumors, there is more healthy non-fibrotic pancreatic tissue left, which might explain the higher incidence of pancreatic fistulas in these patients. Taken together, in our study patients developing POPF grade B/C postoperatively were mainly male and showed more comorbidities including a higher BMI, lower levels of CA 19-9 and less biliary stenting, all factors associated with a higher risk for POPF [9,20,21]. In addition, soft pancreatic gland tissue was associated with development of grade B/C POPF; however, duct diameter < 5 mm was not.

Once pancreatic fistula is evident, it often leads to other complications such as wound infections, intra-abdominal abscess formation, delayed gastric emptying and postpancreatectomy hemorrhage [3,4,21], all of which could be observed in this study. In addition, we observed more reoperations being performed in patients developing POPF. This is in line with our latest findings showing that insufficiency of PG due to leakage of pancreatic fluid and associated POPF is the main reason for reoperations at our center and furthermore, reoperations are associated with a mortality rate of 25% [16]. In particular, another study reported that redo surgery for POPF is even associated with a mortality rate up to 39% [32]. In the case of PPH, interventional angiography was the treatment of choice and only in the case of circulatory instability did patients receive a secondary surgery. Thus, due to these advanced radiologic treatment strategies, operative reinterventions after PD due to PPH have deccreased over the last decades, as we just recently showed [16].

Even though it did not affect the overall mortality rate in our study, but because of the high incidence of POPF and the observed high morbidity with a concomitant prolonged length of the hospital stay, it is of utmost importance to optimize possible treatment strategies. At our center, postoperative management of patients receiving PD is carried out according to our standardized operating procedure protocol for the management of grade B/C POPF (Figure 2). This comprises (a) a conservative treatment strategy if amylase levels are more than three times the normal serum amylase with food restriction and parenteral nutrition as well as the administration of antibiotics and a successive removal of the intraoperatively administered soft drains in the later course; (b) additional drainage placement; (c) endoscopic vacuum-assisted closure therapy (EVT); (d) surgery in the case of early insufficiency of PG within the first three days after surgery or in the case of partial or total dehiscence and its associated complications; or (e) a combination of the afore-mentioned treatment strategies. At our center, the majority of the patients developing grade B/C POPF were able to receive conservative treatment, which indeed led to a shorter length of the hospital stay compared to non-conservative treatment. When combined with interventional drainage placement or EVT, even more patients were able to avoid secondary surgery. Munzo-Bongrad et al., for instance, also reported a successful conservative treatment including additional interventional drainage placement, which was up to 85% [33]. 

An indication for additional non-operative treatment at our center was intra-abdominal abscess formation for drainage placement, either percutaneousely CT-guided or endoscopically via puncture of the stomach. CT-guided drainage was the preferred route for drainage placement since it is less invasive, can be placed in local anesthesia, does not require the expertise of a highly qualified endoscopist and can be considered a valuable tool to prevent secondary surgery [34]. In this study, endoscopic transmural puncture of fluid collections was the treatment route of choice only in one case due to phlegmon of the abdominal wall, and three patients received additional endoscopic drainage in the case of insufficient drainage via percutaneous drains. Both routes of drainage placement seem to be equally efficient [35]; however, in the last few years endocopic drainage placement has been considered a safe alternative [36,37].

If insufficiency of the pancreatic anastomosis, in our case of the PG, was suspected either clinically by monitoring of intraoperatively administered soft drains or by CT scan, endoscopy was initiated and EVT was the treatment of choice. EVT has been widely used for the treatment of gastrointestinal transmual defects, especially in the upper gastrointestinal tract, but also in the rectum [38]. To our knowledge, only two other case reports for the treatment of POPF with EVT have been published [39,40]. At our center, as described previously in a multidisciplinary case serious of eight patients [41], an individually shaped open-pore polyurethane foam sponge or a special open-pore film for very small lesions was connected to the tip of a 14 or 16 French suction catheter and endoscopically placed into the site of the insufficiency of the PG (intracavitary EVT compared to intraluminal EVT, in which the catheter is placed at the perforation site). The suction catheter was transnasally retrieved, fixed to the nose with tape and then connected to a vacuum pump with a continuous negative pressure between 100 and 150 mmHg. Often, EVT is combined with the administration of a nasojejunal tube for enteral feeding. The approach requires the expertise of highly qualified endoscopists, and the sponge/film has to be changed twice per week, each time requiring analgosedation. In general, EVT in the gastrointestinal tract is considered to be safe. The most common side effects are sponge dislocation, minor bleeding and anastomotic strictures, but also major bleeding [38]. Patients mainly complain about nausea, emesis and pain related to the inserted tube. However, successful treatment of insufficiency of PG can be considered a valuable tool—if initiated early enough—to prevent total or partial dehiscence, with dehiscence that is associated with extraluminal hemodynamic relevant bleedings or septic conditions requiring further surgery. 

## 5. Conclusions

Taken together, patients developing POPF grade B/C showed a worse outcome leading to a prolonged length of the hospital stay, but importantly this did not affect the overall mortality rate. Treatment strategies included conservative treatment, drainage placement, EVT and surgery, and a combination of the aforementioned treatments. The majority of the patients were able to receive conservative treatment, which resulted in a shorter length of their hospital stay. 

## Figures and Tables

**Figure 1 biology-12-00178-f001:**
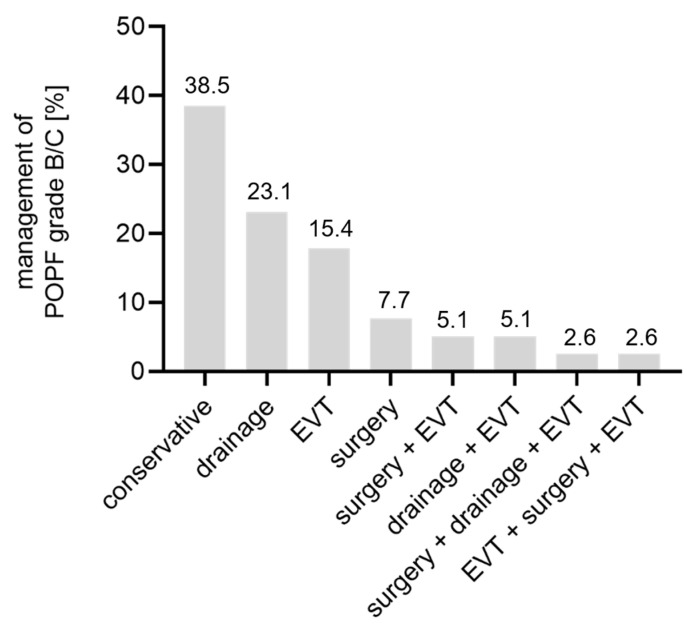
Pancreatic fistula management. EVT = endoscopic vacuum-assisted therapy, POPF = postoperative pancreatic fistula.

**Figure 2 biology-12-00178-f002:**
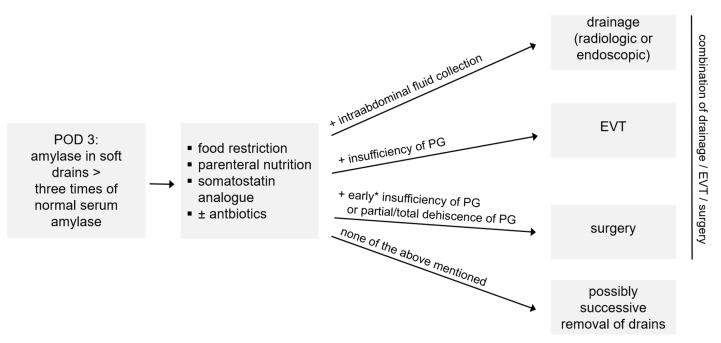
Management of POPF °B/C: how we do it. * within the first 3 days after surgery; EVT = endoscopic vacuum-assisted closure therapy, PG = pancreatogastrostomy, POD = postoperative day.

**Table 1 biology-12-00178-t001:** Clinically relevant PF °B/C—demographic and perioperative data.

	Development of PF °B/C	No Development of PF °B/C	*p*
	*n* = 39	*n* = 161	
age (a)	65 (53–73)	69 (60–76)	0.025
gender female	10 (26%)	74 (46%)	0.021
BMI (kg/m^2^)	27.0 (24.7–31.2)	22.5 (22.0–28.4)	0.002
alcohol abuse	8 (21%)	49 (30%)	0.195
nicotine (active consumption)	9 (23%)	39 (24%)	0.834
weight loss	13 (33%)	93 (58%)	0.010
Charlson Comorbidity Index	2 (1–3)	2 (2–3)	0.014
preoperative diabetes mellitus	9 (23%)	53 (33%)	0.233
preoperative CA 19-9 (U/mL)	14.5 (7.1–49.5)	47.2 (11.2–258.0)	0.007
cholangitis	3 (8%)	15 (9%)	1.000
preoperative biliary drainage	11 (28%)	80 (50%)	0.016
positive intraoperative microbiology	16 (41%)	84 (52%)	0.309
duration of operation (min)	437 (311–491)	403 (333–471)	0.658
blood loss (mL)	400 (300–800)	600 (300–1100)	0.103
transfusions (erythrocyte concentrate)	0 (0–1)	0 (0–2)	0.080
diagnosis malignant	19 (49%)	132 (82%)	≤0.001
positive lymph node histology	11 (28%)	82 (51%)	0.227
extended lymphadenectomy	18 (46%)	86 (53%)	0.394
tumor size (cm)	2.4 (1.2–2.9)	3.0 (2.0–3.9)	0.018
soft pancreas parenchyma	28 (72%)	64 (40%)	≤0.001
pancreatic duct < 5 mm	32 (82%)	119 (74%)	0.350

Data are shown as frequency (%) or median (interquartile range), BMI = body mass index.

**Table 2 biology-12-00178-t002:** Clinically relevant PF °B/C—postoperative outcome/complications.

	Development of PF °B/C	No Development of PF °B/C	*p*
*n*	*n* = 39	*n* = 161	*n*
wound infection (suprafascial)	14 (36%)	21 (13%)	≤0.001
intra-abdominal abscess formation	13 (33%)	24 (15%)	0.009
insufficiency of BDA	5 (13%)	10 (6%)	0.177
insufficiency of DE	6 (15%)	9 (6%)	0.082
PPH grade B/C	13 (33%)	42 (26%)	0.314
DGE grade B/C	18 (46%)	27 (17%)	≤0.001
first day of solid food intake	20 (14–29)	9 (7–14)	≤0.001
intraoperative gastric tube (d)	6 (4–11)	4 (3–6)	0.003
reinsertion of gastric tube	15 (38%)	42 (26%)	0.056
parenteral nutrition (d)	7 (4–15)	0 (0–4)	≤0.001
reoperation	13 (33%)	22 (14%)	0.004
Clavien major (grade III-IV)	33 (85%)	70 (43%)	≤0.001
mortality	4 (10%)	10 (6%)	0.481
stay in intensive care unit (d)	3 (1–4)	1 (1–3)	0.057
length of postoperative stay (d)	28 (22–36)	20 (15–28)	≤0.001

Data are shown as frequency (%), BDA = biliodigestive anastomosis, DE = duodenoenterostomy, PPH = postpancreatectomy hemorrhage, DGE = delayed gastric emptying.

**Table 3 biology-12-00178-t003:** Clinically relevant PF °B/C—risk factors associated with high mortality.

	Odds Ratio	95%-CI	*p*
**univariate**			
preoperative biliary drainage	0.394	0.082–1.124	0.061
positive intraoperative microbiology	0.342	0.103–1.130	0.067
intra-abdominal abscess formation	3.702	1.200–11.421	0.027
PPH grade B/C	3.915	1.291–11.868	0.025
insufficiency of DE	3.932	0.965–16.013	0.076
POPF grade B/C	1.726	0.511–5.824	0.481
**multivariate**			
PPH grade B/C	4.867	1.527–15.512	0.007
intra-abdominal abscess formation	4.492	1.366–14.775	0.013

CI = confidence interval, DE = duodenoenterostomy, POPF = postoperative pancreatic fistula, PPH = postpancreatectomy hemorrhage.

## Data Availability

Our anonymized pancreatic resection database contains sensible data (e.g., date of surgery), with which certain patients could be identified. According to German law and according to the approval of the ethics committee, these data must not be published. Access to the database can be obtained from the corresponding author upon reasonable request.

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
