# Peer review of "Clinically Relevant Pancreatic Fistula after Pancreaticoduodenectomy: How We Do It"

_biology, 2023, doi:10.3390/biology12020178_

Round 1

Reviewer 1 Report

Enderes et al. present a manuscript on postoperative pancreatic fistula and their mode of management.

In all, the manuscript is welI written and scientifically sound. Only, I would like to discuss, whether a detailed patient by patient description of the management is really necessary. I would rather prefer a flow chart for the management to illustrate the decisions, especially in regard to their rather novel approach of EVT in pancreatic fistulas. This novel method should be illustrated a bit more than only a reference to previous work, and its limitations and failures should be mentioned. Then, the novelty of the department's approach ("How we do it") will become considerably better visible. The authors should as well emphasize more that they performed only pancreatogastrostomies (correct?) without risk stratification and that only in this cases EVT is feasible.  

Author Response

First of all, we would like to thank the reviewer for her/his pro bono work and constructive remarks. We tried to meet the suggestions as follows:

Reviewer 1

In all, the manuscript is well written and scientifically sound.

Query 1:

Only, I would like to discuss, whether a detailed patient by patient description of the management is really necessary. I would rather prefer a flow chart for the management to illustrate the decisions, especially in regard to their rather novel approach of EVT in pancreatic fistulas. This novel method should be illustrated a bit more than only a reference to previous work, and its limitations and failures should be mentioned. Then, the novelty of the department's approach ("How we do it") will become considerably better visible.

With the detailed patient-by-patient description we wanted to give the reader the possibility to get further detailed information about the treatment strategies and the postoperative outcome for grade B/C POPF if desired. However, we agree that a flow chart illustrates the treatment strategies more clearly. We would suggest that we offer the patient-by-patient description (Table 4) as supplemental material (new: Supplemental Table 1); in addition, we created a flow chart, as suggested (see Figure 2; discussion page 9) and described the method in somewhat more detail, including possible failures (discussion, page 10).

Query 2:

The authors should as well emphasize more that they performed only pancreatogastrostomies (correct?) without risk stratification and that only in this cases EVT is feasible.

This is correct. In addition to the treatment strategies already mentioned in the method section (page 2: “[…] c) by additional endoscopic vacuum-assisted closure therapy (EVT) in case of insufficiency of PG […]“ to further clarify, we added the words “by default” to the method section on page 2.

Reviewer 2 Report

Authors in the present manuscript aims to investigate possible risk factors for clinical relevant postoperative pancreatic fistula(POPF) grade B/C in their patients population. Altought of some interest there are many points of concern:

1. Fistula B and C must analysed separtely 

2. length of postoperative stay of 20 days in your control group it looks longer. Due you use any ERAS protocol? Please comment

3. 15% of pts in your control group developed intrabdomial post-op abscess. How do you explain? Did you check for amylase in the fluid drained?

4. How many pts underwent neoadjuvant therapy?

5. You report 33% and 26% of PPH grade B/C. No mention of interventional angiography is reported . Please comment on this

Author Response

First of all, we would like to thank the reviewer for her/his pro bono work and constructive remarks. We tried to meet the suggestions as follows:

Reviewer 2

Authors in the present manuscript aims to investigate possible risk factors for clinical relevant postoperative pancreatic fistula(POPF) grade B/C in their patients population. Altought of some interest there are many points of concern:

Query 1:

Fistula B and C must analysed separtely.

The goal of this study was to investigate possible risk factors and to analyze possible treatment strategies for clinically relevant postoperative pancreatic fistula (POPF) grade B/C according to the updated definitions of the International Study Group of Pancreatic Surgery (ISGPS) in 2016. This classification is widely accepted in our opinion. Since with these definitions POPF is classified as either an asymptomatic biochemical leak without a change in clinical management (biochemical leak; BL) or as a symptomatic clinically relevant POPF (grade B/C POPF), which involves a change in clinical management, we deliberately chose to distinguish between patients with a change in clinical management (due to clinically relevant POPF grade B/C) and patients without a change in clinical management (not developing POPF grade B/C). Besides, of the patients developing clinically relevant POPF, only 9 patients showed POPF grade C, which makes solid statistical analysis difficult due to low numbers. However, we agree that clinical management of POPF grade B and C differs, which we pointed out on page 2 in the introduction (“[…] POPF grade B with intraoperatively administered drains either left in place > 3 weeks or requirement of additional percutaneous or transgastric drain placement or for POPF grade C with the need for surgery or occurrence of organ failure“.) and with Table 4 (New: Supplemental Table 1). For further clarification, we have now added a flow chart (Figure 2) for the management of POPF.

Query 2:

Length of postoperative stay of 20 days in your control group it looks longer. Due you use any ERAS protocol? Please comment

The length of postoperative stay in our control group is an average of 20 days compared to an average of 28 days for patients developing POPF grade B/C. The postoperative stay in the control group is comparable to our previous studies (PMID: 34330242, PMID: 34200183, PMID: 35625491, PMID: 36556127) reporting an average length of postoperative stays of 21, 23, 22 and 20 days, respectively, in the control group.

In our department perioperative care is carried out according to our institutional standard operating procedure protocol, which is an ERAS protocol. We have further clarified this in the methods section on page 3.

Query 3:

15% of pts in your control group developed intrabdomial post-op abscess. How do you explain? Did you check for amylase in the fluid drained?

The control group comprises patients who did not develop POPF as well as patients with a biochemical leak, both without a change in clinical management due to POPF. However, even without an evident POPF, patients can develop an intraabdominal abscess due to other reasons (e.g. insufficiency of BDA or DE, encapsulated infectious fluid collection without insufficiency of BDA/DE/PG, bilioma, encapsulated infectious ascites, encapsulated infectious hematoma).

As stated in the methods section, amylase levels were checked in postoperative day 3 (“Amylase levels in abdominal drains were measured on postoperative day 3 by default”).

Query 4:

How many pts underwent neoadjuvant therapy?

13 patients out of the 200 involved in this study received neoadjuvant therapy (6.5%). Of these, 11 patients did not develop POPF, one patient developed POPF BL and another patient developed POPF grade B.

Query 5:

You report 33% and 26% of PPH grade B/C. No mention of interventional angiography is reported . Please comment on this.

We have added a section on interventional angiography to the discussion (page 8/9): “In case of PPH, interventional angiography was the treatment of choice and only in case of circulatory instability patients received a secondary surgery. Thus, for instance, due to these advanced radiologic treatment strategies, operative reinterventions after PD due to PPH have decreased over the last decades, as we just recently showed (PMID: 36556127)“.

Round 2

Reviewer 1 Report

Thank you very much for incorporating my suggestions to further improve the manuscript . I have no more concerns.

Reviewer 2 Report

Authors provided a well point to point responce